# Stillage Waste from Strawberry Spirit Production as a Source of Bioactive Compounds with Antioxidant and Antiproliferative Potential

**DOI:** 10.3390/antiox12020421

**Published:** 2023-02-09

**Authors:** Carmela Spagnuolo, Federica Moccia, Idolo Tedesco, Eva Adabbo, Lucia Panzella, Gian Luigi Russo, Alessandra Napolitano

**Affiliations:** 1Institute of Food Sciences, National Research Council, I-83100 Avellino, Italy; 2Department of Chemical Sciences, University of Naples “Federico II”, I-80126 Napoli, Italy

**Keywords:** strawberry (*Fragaria* spp.), agri-food by-products, antioxidant, antiproliferative effect, fermentation, waste valorization

## Abstract

The production of fruit distillates generates solid residues which are potentially rich in bioactive compounds worthy of valorization and exploitation. We report herein the in vitro antioxidant and antiproliferative properties of an extract obtained from the waste of fermented strawberry distillate production. The main low molecular weight phenolic components of the extract were identified as ellagic acid and p-coumaric acid using spectroscopic and chromatographic analysis. The extract exhibited high antioxidant properties, particularly in the ferric reducing/antioxidant power (FRAP) assay, and a high total phenolic content (TPC). It was also able to induce an antiproliferative effect on different human cancer cell lines. A strong decrease in viability in human promyelocytic leukemia (HL-60) cells through a rapid and massive apoptosis were observed. Moreover, at an early time (<30 min), reactive oxygen species (ROS) production and inactivation of the extracellular signal-regulated kinase (ERK)/mitogen-activated protein kinases (MAPK) pathway were detected. Notably, the antiproliferative activity of the sample was comparable to that observed with an analogous extract prepared from unfermented, fresh strawberries. These results bring new opportunities for the valorization of fruit distillery by-products as low-cost resources for the design of bioactive formulations of comparable value to that from fresh food.

## 1. Introduction

Accordingly to FAO estimation, 221 million tons of food are lost and wasted every year in the production and retail chains [1]. To counteract this global issue, several strategies have been and are currently being pursued, among which is the valorization of agri-food wastes and by-products as sources of bioactive compounds to be used, e.g., as functional food ingredients or nutraceuticals. Plant-based by-products are in general rich in functional compounds, including polysaccharides, vitamins, dietary fibers, oils, and phenolic compounds [2,3,4,5]. These latter, in particular, are well-known for their healthy functions, including antioxidant, antimicrobial, antiproliferative, anti-inflammatory and neuroprotective properties [6,7,8,9]; therefore, the recovery of phenolic compounds from agro-industrial by-products is today a cutting-edge issue within the field of circular economic systems. In this context, the exploitation of biotechnological methodologies allowing for the enrichment of bioactives is one of the most pursued approaches [10,11,12,13,14].

Fruit distillates are among the oldest liqueurs in the world and each country has its own typical regional ones. They arise from the ingenuity of the ancient farmers, who recovered the waste fruits, i.e., the ones too ripe or too spoiled to be eaten, and, after allowing for spontaneous fermentation, distilled them. Even today the production of fruit distillates involves a spontaneous fermentation process of homogenized fruits followed by distillation, and this procedure generates huge amounts of solid residues potentially rich in bioactive compounds that could be further exploited and valorized.

Among fruits, the strawberry (*Fragaria* spp.) has been described as a rich source of bioactive compounds with potential health effects [15,16]. In the agri-food industry, this fruit is widely employed to produce various processed products, e.g., juice, jam, and also distillates, giving rise, however, to a quantity of waste raw material that has not yet been fully exploited. Indeed, only a few recent papers have focused on the possible valorization of strawberry extrudate or strawberry pomace for the recovery of volatile fatty acids [17] and bioactive compounds [18,19,20], or of strawberry seeds for the preparation of activated carbon [21]. On the other hand, several in vitro and in vivo studies have highlighted the health properties of strawberries and their extracts, such as the preventive effect against inflammation, cardiovascular diseases, metabolic syndrome, cancers, and neurological diseases [15,22,23], mostly related to the presence of phenolic compounds such as flavonoids, mainly anthocyanins (pelargonidin-3-glucoside and to a lesser extent cyanidin-3-glucoside), ellagitannins, and phenolic acids [15,24]. The anti-cancer effect of strawberry-derived phenolic extracts has been mainly correlated to the ability of phenolic compounds to reduce the levels of reactive oxygen species (ROS) and oxidative DNA damage, to detoxify carcinogens, and to inhibit cancer cell proliferation inducing apoptosis and cell-cycle arrest modulating key pathways such as the activator protein-1, NF-κB, PI_3_K, and inhibiting the Wnt signaling [22,25,26].

In this context, we report herein the in vitro antioxidant and antiproliferative properties of an extract prepared from the waste generated by the production of a fermented strawberry distillate, hereafter indicated as fermented strawberry extract (FSE), with the aim to provide preliminary data for a full exploitation of this easily available material as a source of bioactive compounds.

## 2. Materials and Methods

### 2.1. Reagents

HT-29 cell line was purchased from LGC Standards (Sesto San Giovanni, Milan, Italy); promyelocytic leukemia cell line (HL-60) was purchased from Merck Life Science (Milan, Italy); two leukemia T cell lines (HPB-ALL and Jurkat) were available at the Institute of Food Science, Avellino, Italy; a hepatoma cell line (HepG2) and a human osteosarcoma cell line (U2Os) were purchased from ATCC (Manassas, VA, USA). Roswell Park Medium Institute (RPMI); Dulbecco’s Modified Eagle’s Medium (DMEM) (Euroclone, Milan, Italy), fetal bovine serum, L-glutamine 200 mM, penicillin 5000 IU mL^−1^, streptomycin 5000 μg mL^−1^, and phosphate-buffered saline (PBS) were purchased from Thermo Fisher Scientific/Life Technologies (Milan, Italy); crystal violet, trypan blue solution (0.4% *v*/*v*), 2,2-diphenyl-1-picrylhydrazyl (DPPH), 2,4,6-tripyridyl-s-triazine (TPTZ), Folin–Ciocalteu′s phenol reagent, ellagic acid (EA), *p*-coumaric acid, catechin, gallic acid, Trolox, vanillin, propidium iodide, dithiothreitol, phenylmethylsulfonylfluoride (PMSF), pepstatin A, aprotinin, leupeptin, and 2-morpholinoethanesulfonic acid monohydrate (MES) were from Merck Life Science (Milan, Italy); 2′,7′-dichlorofluorescein diacetate (DCFH-DA), dihydroethidium (DHE) were from Thermo Fisher Scientific/Molecular Probe; ferric chloride, aluminum chloride, formalin were from Carlo Erba (Milan, Italy); Annexin V, and caspase-3 and 9 kits were from Enzo Life Sciences (Lausen, Switzerland). All reagents used were of pure, analytical grade.

### 2.2. Preparation of FSE 

Strawberry distillation residues were provided by Berolà Distillati (Portico di Caserta, Caserta, Italy), together with fresh fruits of the same batch not subjected to fermentation, and stored at −25 °C. After thawing, drying, and grinding, 0.3 g of material was kept under vigorous magnetic stirring in 3 mL of dimethyl sulfoxide (DMSO). After 1 h the mixture was centrifuged at 5478× *g* (Thermo Scientific Heraeus Biofuge Primo R, Rodano, Italy) for 20 min at room temperature and the supernatant (FSE) was stored at −25 °C until further analysis. When required an analogous extract was prepared starting from fresh strawberry (strawberry extract, SE).

### 2.3. Chromatographic and Spectral Characterization of FSE 

HPLC analyses were performed at room temperature on an Agilent (Cernusco sul Naviglio, Milan, Italy) instrument equipped with a UV-Vis detector, using a Phenomenex (Castel Maggiore, Bologna, Italy) Sphereclone ODS column (250 × 4.60 mm, 5 µm) at a flow rate of 1.0 mL min^−1^. The gradient elution was as follows: 0.1% formic acid (solvent A)/methanol (solvent B): 5% B, 0–10 min; from 5 to 80% B, 10–47.5 min; the detection wavelength was set at 254 nm. LC-MS analyses were run on an Agilent LC-MS ESI-TOF 1260/6230DA instrument (Cernusco sul Naviglio, Milan, Italy) operating in positive ionization mode; the following conditions were adopted: capillary voltage 3500 V; drying gas (nitrogen) 5 L min^−1^, 325 °C; fragmentor voltage 175 V; nebulizer pressure 35 psig; an Agilent Eclipse Plus ODS column (150 × 4.6 mm, 5 µm) (Cernusco sul Naviglio, Milan, Italy) was used, with the same eluant as above at a flow rate of 0.4 mL min^−1^. UV-Vis spectra were recorded on a HewlettPackard8453 Agilent spectrophotometer. For NMR analysis, the extraction was performed in DMSO-*d*_6_ starting from 0.1 g of material. ^1^H NMR spectra were recorded at 400 MHz on Bruker (Milan, Italy) instruments.

### 2.4. DPPH Assay 

FSE (2.5–40 µL) was added to a 0.2 mM ethanolic solution of DPPH. After 10 min under stirring at room temperature the absorbance at 515 nm was measured [27]. Trolox was used as a reference antioxidant. Experiments were run in triplicate. 

### 2.5. Ferric Reducing/Antioxidant Power (FRAP) Assay 

FSE (0.66–5 µL) was added to 0.3 M acetate buffer (pH 3.6) containing 1.7 mM FeCl_3_ and 0.83 mM TPTZ [28]. After 10 min under stirring at room temperature the absorbance of the solution at 593 nm was measured. Trolox was used as a reference antioxidant. Experiments were run in triplicate.

### 2.6. Determination of Total Phenolic Content (TPC) 

FSE (2.5–30 µL) was added to a solution consisting of Folin–Ciocalteu reagent, 75 gL^−1^ sodium carbonate, and water in a 1:3:14 *v*/*v*/*v* ratio [29]. After 30 min incubation at 40 °C, the absorbance at 765 nm was measured. Gallic acid was used as a reference compound. Experiments were run in triplicate.

### 2.7. Determination of Total Flavonoid Content (TFC)

FSE (50–200 µL) was added to water up to a total volume of 600 µL and then mixed with 600 µL of a 2% *w*/*w* aluminum chloride solution in water [30]. After mixing, the solution was incubated for 60 min at room temperature. The absorbance of the reaction mixtures was measured at 420 nm. Quercetin was used as a standard. Experiments were run in triplicate.

### 2.8. Determination of Condensed Tannin Content (CTC) 

FSE (25–240 µL) was added to 1 mL of a 1% *w*/*v* vanillin solution in methanol [31]. One mL of 9 M HCl was added, and the mixture was incubated at 30 °C for 10 min. Finally, the absorbance at 505 nm was measured. Catechin was used as a reference compound. Experiments were run in triplicate.

### 2.9. Cell Culture and Viability Assays 

HL-60, HPB-ALL, and Jurkat cells were cultured in RPMI supplemented with 10% fetal bovine serum, 1% penicillin/streptomycin, and 1% l-glutamine. HT29, HepG2 and U2Os cells were cultured in DMEM supplemented with 10% fetal bovine serum, 1% l-glutamine, 1% penicillin/streptomycin, and 1% NEAA at 37 °C, in a humidified atmosphere containing 5% CO_2_. Cell viability in HT29, HepG2, and U2Os cells was assayed using crystal violet staining. Briefly, cells were cultured at a density of 8 × 10^4^ mL^−1^ in 48-well plates and treated with different concentrations of the extracts at the indicated times. After stimulation, cells were fixed with 10% formalin for 10 min and washed with water before the addition of crystal violet (0.05% *w*/*v*) for 30 min. Finally, 10% acetic acid was added and absorbance was spectrophotometrically measured at 590 nm. The quantity of adsorbed dye was proportional to the number of living cells. Cell viability in HL-60, HPB-ALL, and Jurkat cells was determined by CyQuant kit (Thermo Fisher Scientific/Life Technologies) as described [32]. CyQuant is a cell-permeant fluorescent DNA-binding dye (CyQuant nuclear stain) used in combination with a background suppressor able to quantify cell proliferation and cytotoxicity. CyQuant mixture was added to the culture medium for 1 h at 37 °C, and fluorescence was measured at an excitation wavelength of 485 nm and emission of 530 nm in a multiplate reader (Synergy HT, BioTek, Milan, Italy).

### 2.10. Apoptotic Assays 

To verify the induction of apoptosis, three different assays were used: apoptotic bodies detection, Annexin V exposure, and caspase-9 and -3 enzymatic activities. 

*Apoptotic bodies detection:* The CyQuant kit was used to visualize cellular nuclei. Briefly, cells were stimulated for 24 h with the extracts and then incubated with CyQuant mixture. Subsequently, the apoptotic nuclei were visualized using a fluorescence microscope (Zeiss Axiovert 200, Milan, Italy) and photographed in a FITC filter.

*Annexin V detection*: Phosphatidylserine exposure was measured using the binding of fluorescein-isothiocyanate-labelled (FITC) Annexin V to phosphatidylserine (PS), as indicated in the manufacturer’s protocol (Miltenyi Biotec, Bologna, Italy). Briefly, after 15 h of treatment, HL-60 cells (0.15 × 10^6^ mL^−1^) were washed and suspended in 100 μL of binding buffer. Cells were incubated with FITC Annexin V (10 μL) in the dark at room temperature and, after centrifugation, were re-suspended in 500 μL of binding buffer and 25 μg mL^−1^ of propidium iodide immediately prior to analysis with flow cytometry. Low-fluorescence debris and necrotic cells were gated out before the measurements. About 10,000 events were collected and data were analyzed using CellQuest software. 

*Caspases assay*: For caspase-9 and -3 enzymatic activities, HL-60 cells (0.15 × 10^6^ mL^−1^) were incubated for 6 h, as described above. At the end of incubation, cells were washed twice in PBS and suspended in lysis buffer (10 mM HEPES, pH 7.4; 2 mM ethylenediaminetetraacetic acid; 0.1% 3-((3-cholamidopropyl) dimethylammonium)-1-propanesulfonate; 5 mM dithiothreitol; 1 mM phenylmethylsulfonylfluoride; 10 μg mL^−1^ pepstatin-A; 10 μg mL^−1^ aprotinin; 20 μg mL^−1^ leupeptin). Following measurement of protein concentration, cell extracts were added with reaction buffer and the respective conjugated amino-4-trifluoromethyl coumarin (AFC) substrates, benzyloxycarbonyl-Asp(OMe)-Glu(OMe)-Val-Asp(OMe)-AFC(ZDEVD-AFC) for caspase-3 and LEHD-AFC for caspase-9 (carbobenzoxy-Asp-Glu-Val-Asp and Leu-Glu-Hys-Asp-AFC) before incubation at 37 °C for 30 min. Upon proteolytic cleavage of the substrates by the different caspases, the free fluorochrome AFC was detected with excitation and emission setting of 395 ± 20 and 530 ± 20 nm, respectively (Synergy HT microplate reader, BioTek). To quantify the enzymatic activities, an AFC standard curve was determined. Caspase-specific activities were calculated as nmoles of AFC produced per minute per mg of proteins at 37 °C at saturating substrate concentrations (50 μM). Fold increase in caspase-3 and -9 activities was determined by direct comparison with the level of DMSO-treated control cells. 

### 2.11. Immunoblottings 

HL-60 cells were incubated with extracts as indicated above and, at the end of incubation, cells were lysed using a lysis buffer containing protease and phosphatase inhibitors, as reported [33]. Following protein concentration determination [34], 30 μg of the total protein lysates were loaded on a 4–12% precast gel (Novex Bis-Tris precast gel 4–12%, Life Technologies, Monza, Italy) using MES buffer. The immunoblots were performed following standard procedures, using as primary antibodies anti-Caspase-3, anti-p42/44, anti-phosphoBcl-2(Ser70), anti-Bcl-2 (Cell Signaling Technology), anti-ERK1, anti-PARP (Santa Cruz Biotechnology, Heidelberg, Germany) and anti-α-tubulin (Merck Life Science, Milan, Italy). PVDF membranes were finally incubated with horseradish peroxidase-linked secondary antibody raised against mice, and immunoblots developed using the ECL Plus Western blotting detection system kit (GE Healthcare, Milan, Italy). Band intensities were quantified by measuring optical density on a Gel Doc 2000 Apparatus and multi-analyst software (Bio-Rad Laboratories, Milan, Italy).

### 2.12. Intracellular Oxidative Stress Measurement 

HL-60 cells, 1 × 10^4^ mL^−1^ in 96-well plates, were incubated for 30 min with 10 μM DCFH-DA, a ROS-sensitive probe, or with 10 μM superoxide probe DHE, and subsequently stimulated with the extracts as indicated in Section 3. After incubation, the cells were washed twice with PBS before the determination of fluorescence by the multiplate reader with an excitation and emission setting of 485 nm and 530 nm, respectively, for DCFH-DA or with excitation of 400 nm and emission of 490 nm for DHE, expressed as the percentage of fluorescence with respect to untreated cells.

### 2.13. Statistical Analysis 

Data are expressed as the mean ± standard deviation (SD). To evaluate the significance, data were analyzed using Student’s *t*-test of the single treatment vs. control because the data were continuous, there was homogeneity in the variance, the distribution was approximately normal, and the samples were independent.

## 3. Results and Discussion

### 3.1. Structural and Antioxidant Property Characterization of FSE 

To extract phenolic compounds from the fermented strawberry distillate production waste, DMSO was chosen as the solvent based on its ability to dissolve a wide range of most polar and non-polar natural phenolic compounds and on its compatibility with cellular assays. HPLC analysis of the extract after proper dilution in methanol (Figure 1) showed the presence of a main compound eluted at ca. 35 min, which was identified as EA based on LC-MS analysis and on the comparison of the chromatographic properties with those of an authentic standard. Furthermore, the elutographic profile recorded at 320 nm showed, besides EA, very low amounts of a compound eluted at ca. 28 min, identified as p-coumaric acid. The quantitative analysis allowed us to determine the concentrations of EA and p-coumaric acid in FSE as 0.170 ± 0.003 mg mL^−1^ and 0.004 ± 0.001 mg mL^−1^, respectively. Although the occurrence of both compounds has previously been reported in strawberries [35,36], several papers in the literature have described the increased formation of EA as well as of p-coumaric acid in fermented plant products, likely as the result of hydrolytic processes brought about by microorganisms [12,14,37,38,39]. Indeed, none of these compounds was found to be present in the DMSO extract of fresh strawberries of the same variety (strawberry extract, SE) (Appendix A), suggesting their actual generation during the fermentation process. 

The UV-Vis spectrum of FSE showed a maximum at around 360 nm (Figure 2a), as expected based on the presence of EA. Notably, no other low molecular weight phenolic compounds were found to be present in FSE, as evident also from 1H NMR analysis. In fact, the aromatic region of the spectrum (Figure 2b) was characterized by the presence, besides the signal due to EA at ca. 7.4 ppm, of very broad signals likely indicative of the presence of heterogeneous phenolic polymers. 

Based on the antioxidant properties previously reported for EA [40] and on the amount of EA present in the extract, Trolox equivalent (eqs) values of 0.0144 mg/mL and 0.17 mg/mL should have been determined for FSE in the DPPH and FRAP assay, respectively. As reported in Table 1, values 2.6- and 7.6-fold higher compared to the expected ones were observed for FSE in the two assays. This result indicates the presence of significant amounts of additional phenolic compounds in the extract. Indeed, the Folin–Ciocâlteu and the vanillin-HCl assays indicated relatively high TPC and CTC, whereas a very low TFC was determined (Table 1). Notably, as far as the antioxidant properties are concerned, FSE was found to be particularly active in the FRAP assay rather than in the DPPH assay, pointing to an electron donor ability instead of an H-donor ability as the main determinant of the reducing properties observed [41]. 

### 3.2. Effects of FSE on Cell Viability of Different Tumor Cell Lines

To verify the biological potential of FSE, its antiproliferative activity was investigated on a panel of different in vitro cellular models, represented by solid tumor and leukemic cell lines differently sensible to cell death, namely: a promyelocytic leukemia cell line (HL-60), two leukemia T cell lines (HPB-ALL and Jurkat), a human colon cancer cell line (HT-29), a hepatoma cell line (HepG2), and a human osteosarcoma cell line (U2Os). Cells were treated for 24 h with the extract at 25 and 50 μM gallic acid equivalent (eqGA) and, as reported in Figure 3a, the treatment significantly reduced the viability of the three leukemic cell lines (HL-60, HPB-ALL and Jurkat cells), while modest or null effects were observed in the solid tumor cell lines (HT-29, HepG2, and U2Os cells). Notably, the effects induced by FSE (Figure 3a) on cell viability were found to be comparable to those exerted by SE (Figure 3b).

Starting from these data, an in-depth study was conducted on HL-60, the cellular model that proved more sensitive to the treatment. After stimulating cells for 24 h within a range of concentrations corresponding to 12.5–50 μM eqGA, a rapid and strong response to both FSE and SE was observed, with about a 70% decrease in cell viability at the concentration of 50 μM eqGA (Figure 4a). The EC_50_ values calculated for FSE were 36.7 ± 6.4 μM eqGA and, similarly, 30.7 ± 3.1 μM eqGA for SE. Moreover, HL-60 cells were also differentiated with PMA to mimic the healthy counterpart of the selected cell model. As reported in Figure 4b, in PMA-differentiated HL-60, the extracts did not show any cytotoxic effect at the used concentrations, thus suggesting a specific action of FSE against cancer cells.

The similar antiproliferative effect generated by FSE and SE rules out any detrimental effects of the fermentation/distillation processes on the well-known biological activity of strawberries [15,16] and could be attributed to the action of similar bioactive compounds present in both extracts. This hypothesis was also supported by data obtained from treating HL-60 cells with the two main phenolic components identified in FSE, i.e., EA and p-coumaric acid, which did not induce any significant effect on cell viability (data not shown). 

Phenolic compounds can generate hydrogen peroxide when interacting with culture media components [42]. To avoid this artifactual phenomenon, FSE was incubated with DMEM and RPMI medium in control experiments. No significant amount of hydrogen peroxide was generated, excluding interferences with cell growth (data not shown).

### 3.3. Pro-Apoptotic Effects of FSE on HL-60 Cell Line 

Observation under the microscope evidenced the presence of numerous apoptotic bodies after nuclear staining of cells treated for 24 h with both FSE and SE (Figure 5a). To confirm that the measured reduction in cell viability induced by both extracts was due to the induction of apoptotic cell death, specific assays were performed. PS externalization, an early signal of induction of the apoptotic process, was assessed through the cytofluorimetric assay using the binding of Annexin V. FSE and SE efficiently and significantly induced apoptosis with a strong increase in Annexin V positivity of approximately 30-fold and 25-fold, respectively, with respect to untreated cells (Figure 5b). The activation of caspases-9 and -3 was then assessed. The former is the initiator caspase in the intrinsic apoptotic pathway that proceeds with the activation of effector caspases, such as caspase-3, responsible for the cleavage of substrates such as poly(ADP-ribose) polymerase (PARP) [43]. As reported in Figure 5c,d, a significant increase in caspase-3 and -9 activity was detected compared to untreated cells after 5 h of treatment with both extracts. 

The immunoblottings in Figure 5e further confirm the activation of caspase-3, since the total protein expression was strongly reduced after 6 h of treatment with both FSE and SE, and in parallel, a significant proteolytic cleavage of PARP was triggered.

### 3.4. Pro-Oxidant Activity of FSE in HL-60 Cells 

Considering the critical role that ROS exert on cellular processes such as signaling and cell death [44], the impact of FSE on the redox status of HL-60 cells was then assessed by measuring the levels of intracellular peroxides and superoxide anions using two specific probes, CM-DCFDA and DHE, respectively. As shown in Figure 6a, in HL-60 cells, FSE at 50 μMeqGA induced a significant increase in intracellular ROS (about 40%) after 30 min and 1 h of incubation. Concerning the intracellular production of superoxide, a slight but significant increment of about 10% was observed after treatment (Figure 6b). Similar results were obtained with SE treatment. Although the DPPH and FRAP assays indicated antioxidant properties for FSE, it is probably pro-oxidant o-quinones and/or other oxidation products of the phenolic compounds are responsible for the modulation of the redox balance at the cellular level. This effect, widely described in the literature [8], can occur under specific conditions depending, for example, on the concentration and the nature of surrounding molecules. These results suggest that the phenolic extracts under investigation may probably trigger signaling networks that culminate in apoptotic cell death through a pro-oxidant effect.

### 3.5. Down-Regulation of the Anti-Apoptotic Protein Bcl-2 and Inactivation of the ERK/MAPK Pathway by FSE 

The Bcl-2 protein and the ERK/MAPK pathway are key actors involved in triggering and regulating the apoptotic process [45,46]. The immunoblot in Figure 7 shows that FSE and SE are able to inactivate the ERK/MAPK pathway (by about 1.9- and 1.5-fold, respectively) in HL-60 cells after only 10 min of incubation. Moreover, a reduction of about 1.4- and 1.9-fold of the phosphorylated form of Bcl2 (pBcl-2^Ser70^), the active form of the protein, was also observed, probably as a consequence of extracellular signal-regulated kinase (ERK) inhibition. In fact, the anti-apoptotic protein Bcl-2 is a key factor in resistance to apoptosis and is considered one of the hallmarks of cancer. The post-translational modifications of Bcl-2 family members are essential in anti-apoptotic activity. In particular, the phosphorylation of Bcl-2 at serine-70, mainly by the mitogen-activated protein (MAP) kinases such as ERK, c-Jun N-terminal kinase, and p38, has been shown to increase sequestration of the pro-apoptotic protein BAX [47,48,49,50], resulting in resistance to apoptosis. These data may suggest that bioactive compounds present in FSE rapidly act on the ERK signaling with a possible consequent reduction in Bcl2 post-translational modification. 

As summarized in Figure 7b, the antiproliferative effect induced by FSE and SE can be the outcome of two concomitant processes generated through a double hit: the pro-oxidant action and the inactivation of the anti-apoptotic protein Bcl-2 mediated by ERK/MAPK pathway. In eukaryotic cells, a complex system regulates the production and the response to ROS that are involved in several aspects of cell response from signaling to death. ROS accumulation can provoke several forms of cell death (apoptosis, senescence, autophagy, ferroptosis), limiting tumor development [44]. Different mechanisms controlling ROS have been shown to play a role in tumorigenesis, including GSH and NADPH production, the transcription factor nuclear factor erythroid 2-related factor 2 (NRF2), and others [44]. On the other hand, regarding the MAPKs, which play a pivotal role in converting extracellular stimuli into cellular responses (cell growth, proliferation, differentiation, migration, and cell death), JNK and p38 MAPK are activated mainly following cell exposure to stress (physical, chemical, and biological stimulation), whereas ERK1/2 cascades are triggered mostly by cell growth factor-stimulated signaling [46]. The present study suggests that FSE acts on multiple targets that independently converge on cell death. Moreover, considering that, the main cause of sustained ERK activation is the presence of ROS during the cell death program [51], and that in the JNK/p38 MAPK cascades, ROS, acting as second messengers, regulate MAPKs activation in a positive circuit [52,53], a cross-talk between changes in ROS concentration and MAPKs activation may exist leading to irreversible apoptosis.

## 4. Conclusions

In conclusion, the present work aimed to test the biological potential of FSE as an easily available and green source of bioactive compounds. The obtained results are of particular interest since they demonstrate an antiproliferative activity for this waste-derived product comparable to that exhibited by an analogous extract from fresh strawberries. It will be interesting in the near future to clarify the details of the molecular mechanism responsible for the antiproliferative activity and to confirm these results in vivo also in light of the ever-growing interest in the prevention and treatment of tumor pathologies. Therefore, the results here presented open interesting new perspectives toward the valorization of fruit distillery by-products for the design of bioactive formulations of comparable value to that produced from fresh food, allowing for a rational exploitation of low-cost resources in a circular economy. Moreover, the enrichment in EA and *p*-coumaric acid resulting from the strawberry fermentation process could function to obtain other types of biological effects such as anti-inflammatory and neuroprotective activity, as supported by data in the literature showing the potential activities of these compounds ([54,55]). In addition, based on the antioxidant properties emerging from the chemical assays, other possible applications of FSE could be explored such as a natural antioxidant alternative to synthetic preservatives in the food industry.

## Figures and Tables

**Figure 1 antioxidants-12-00421-f001:**
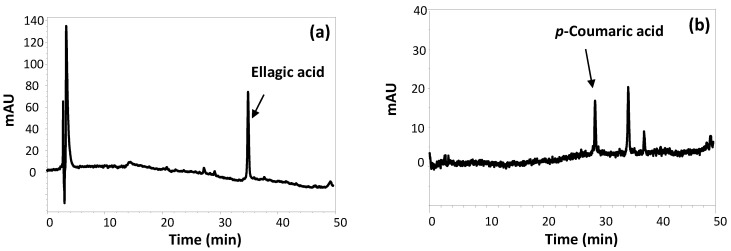
HPLC profile of FSE (diluted 1:5 *v*/*v* in methanol), recorded at (**a**) 254 nm and (**b**) 320 nm.

**Figure 2 antioxidants-12-00421-f002:**
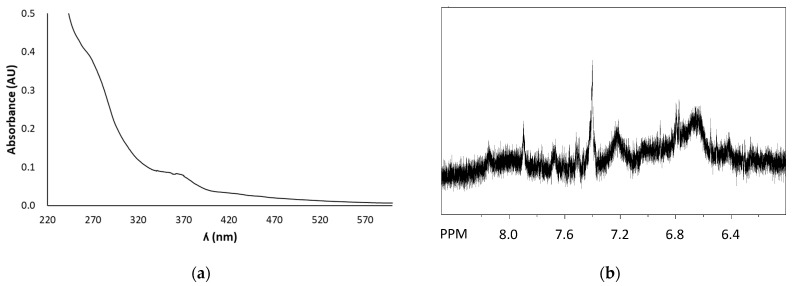
(**a**) UV-Vis spectrum (after dilution 1:200 *v*/*v* in methanol) and (**b**) selected region of ^1^H NMR spectrum of FSE.

**Figure 3 antioxidants-12-00421-f003:**
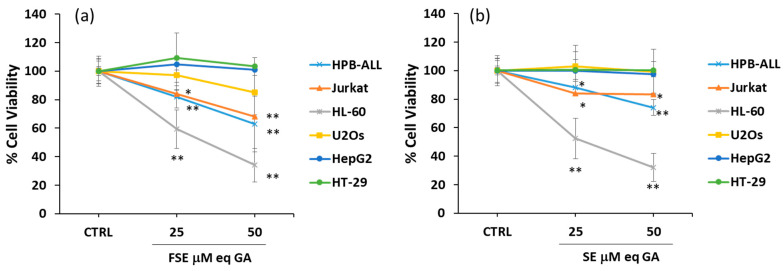
FSE effect on cell viability in different cell lines. HPB-ALL, Jurkat, HL-60, U2Os, HepG2, HT-29 cell lines were treated for 24 h with (**a**) FSE and (**b**) SE at the indicated concentrations. Cell viability was assessed using a crystal violet and CyQuant assay as reported in Materials and Methods section. Bar graphs represent the mean of three experiments (±SD). Symbols indicate significance: *p* < 0.05 (*) and *p* < 0.005 (**) respect to control (CTRL) (DMSO treated cells).

**Figure 4 antioxidants-12-00421-f004:**
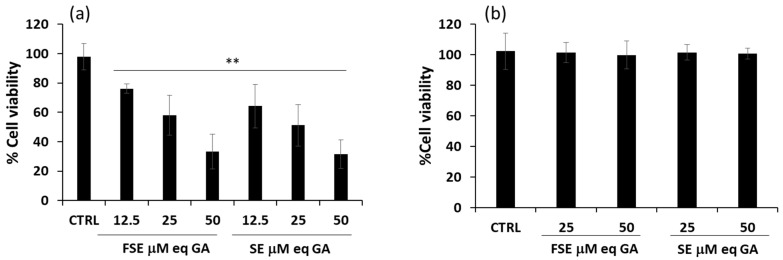
FSE effect on cell viability in (**a**) HL-60 and (**b**) differentiated HL-60 cell lines. Cells were treated for 24 h with FSE or SE at the indicated concentrations. Cell viability was assessed using a CyQuant assay as reported in Materials and Methods section. Bar graphs represent the mean of three experiments (±SD). Symbols indicate significance: *p* < 0.005 (**) respect to CTRL (DMSO-treated cells).

**Figure 5 antioxidants-12-00421-f005:**
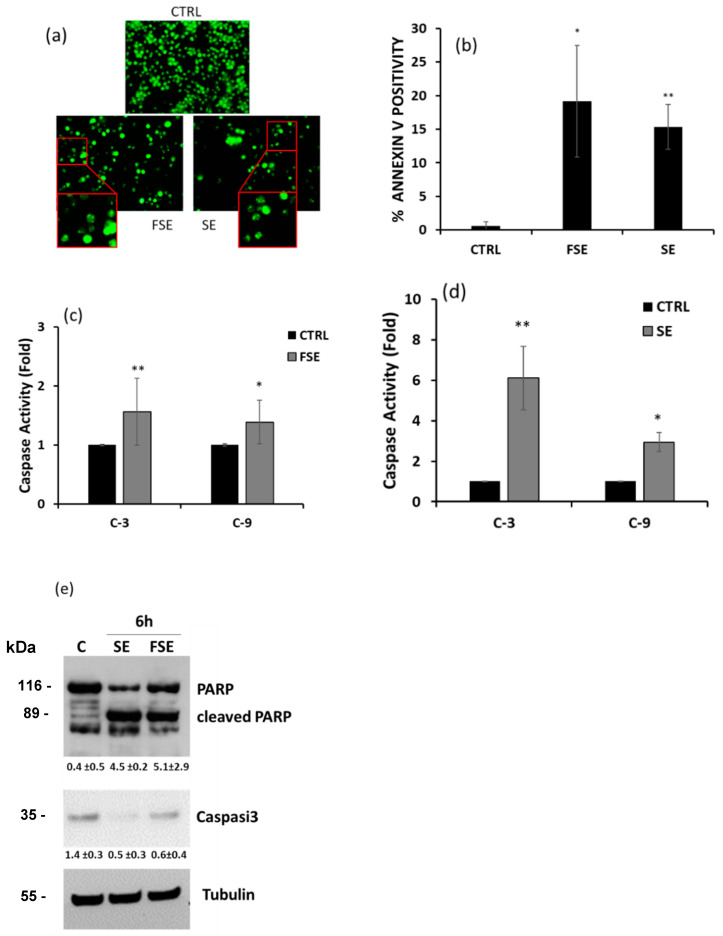
FSE induces apoptosis in HL-60 cells. (**a**) Representative images of cells untreated (top) and treated with FSE and SE (bottom) stained with CyQuant. Cells were visualized using fluorescent microscopy and photographed in FITC filter with 200× magnification. (**b**) PS externalization was evaluated by treating cells for 15 h and measuring the percent of Annexin-V positivity using cytofluorimetry. The proteolytic activity of caspase-3 and -9 (nmol AFC/min/μg protein) induced by (**c**) FSE and (**d**) SE at 25 μM eqGA was measured after 6 h of treatment. Bar graphs represent means ± SD derived from three separate experiments. In panel (**e**) the immunoblots showing PARP cleavage and the reduction in pro-Caspase-3 in HL-60 cells are reported. Numbers between panels indicate the densitometric analysis ± SD of three independent experiments. Symbols indicate significance: *p* < 0.05 (*) and *p* < 0.005 (**) respect to CTRL.

**Figure 6 antioxidants-12-00421-f006:**
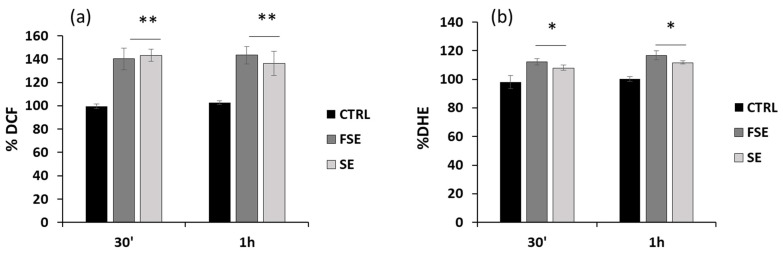
Pro-oxidant effect of FSE in HL-60 cells. Cells were treated with 50 μM eqGA of FSE and SE at the indicated time and intracellular (**a**) ROS and (**b**) superoxide were measured as DCF and DHE fluorescence, respectively. Bar graphs represent means of three separate experiments. ± SD Symbols indicate significance: *p* < 0.05 (*) and *p* < 0.005 (**) respect to CTRL.

**Figure 7 antioxidants-12-00421-f007:**
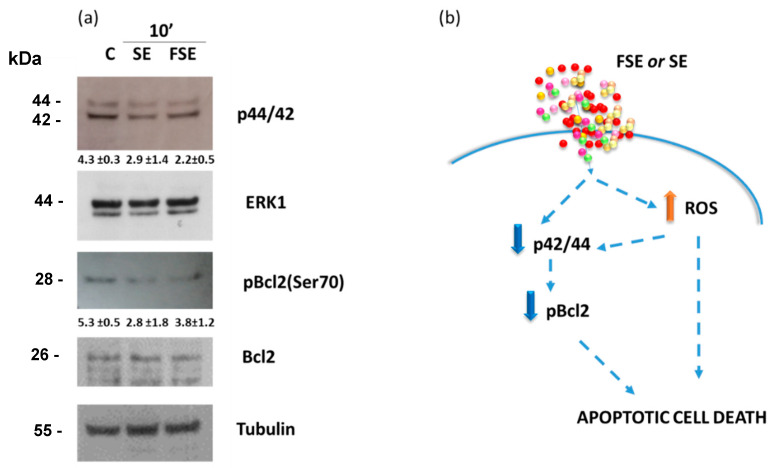
FSE modulation of ERK/MAPK pathway and phosphoBcl-2(Ser70). (**a**) Immunoblot analysis of p44/42-ERK1 and phosphoBcl2(Ser70)-Bcl2 expression in HL-60 cells treated with 25 μM eqGA of FSE or SE for 10 min. Densitometric analysis (numbers between panels ± SD) was obtained by normalizing the expression of p44/42 with ERK1 and tubulin, and the expression of pBcl2(Ser70) with Bcl2 and tubulin. (**b**) Schematic representation of the possible mechanism of the action exerted by FSE.

**Table 1 antioxidants-12-00421-t001:** Antioxidant properties and phenolic content of FSE. Reported are the mean ± SD values of at least three experiments.

Trolox Eqs ^a^ (DPPH Assay)	Trolox Eqs ^a^ (FRAP Assay)	TPC ^b^	TFC ^c^	CTC ^d^
0.0375 ± 0.0009	1.30 ± 0.02	1.40 ± 0.01	0.022 ± 0.002	0.223 ± 0.006

^a^ mg of Trolox eqs/mL of sample. ^b^ mg of gallic acid eqs/mL of sample. ^c^ mg of quercetin eqs/mL of sample. ^d^ mg of catechin eqs/mL of sample.

## Data Availability

Data will be made available on request.

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
