# Peer review of "Stillage Waste from Strawberry Spirit Production as a Source of Bioactive Compounds with Antioxidant and Antiproliferative Potential"

_antioxidants, 2023, doi:10.3390/antiox12020421_

Round 1
Reviewer 1 Report
The article “Stillage waste from strawberry spirit production as a source of bioactive compounds with antioxidant and anti-proliferative potential” by Carmela Spagnuolo et al. is structured following the classic model for this type of material (Research Article) comprising four parts: Introduction, Materials and Methods, Results and Discussion, and Conclusions. The list of bibliographic references is adequate; the documentation is appropriate regarding the titles consulted.
In my opinion, the work presented is weak. Many of the results are not well justified and are inaccurate. Requires more robust graphics with better-explained results. Here are some of the aspects that authors should pay attention to:
_Line 102 - In subpoint 2.3. it is necessary to enter the temperature at which the HPLC tests were carried out.
_Line 159 - In subpoint 2.10. authors must briefly describe the assay "apoptotic bodies detection". Please verify.
_Line 214 - In subpoint 3.1. the authors must present the chromatogram referring to the SE to validate their statements. I think the authors also have to make considerations about the baseline of figure 1(b). Please complete.
_Line 240-243 - I ask the authors to better explain the sentence in these lines. Please explain what you mean by the phrase "TPC and antioxidant properties about 3.7-, 2.6- and 7.6-fold higher than that expected based on the amount of EA present in the extract". This part of the job is confusing. Please check.
Author Response
We thank R1 for the careful revision of our manuscript. Please see the attachment for the point-by-point response to the reviewer’s comments

Reviewer 2 Report
This work is interesting and useful not only from the scientific-technological but also from the industrial point of view. This paper studies a co-product that is not normally used and represents an environmental problem. The implications for health, particularly the inhibition of cancer cell growth, make it the work, a priori, a very interesting field of work and study, not only from the nutritional point of view but also its potential use in pharmacology.
Only minor considerations must be taken into account, previous to its publication.
Line 14, 257. Latin words (in vitro) must be in italics.
Line 214 Please check and delete the point (.) before Structural
Only, the authors must give a better explanation of why they use the Student’s t-test as statistical analysis.
Author Response
We thank R3 for the careful revision of our manuscript. Please see the attachment for the point-by-point response to the reviewer’s comments

Round 2
Reviewer 1 Report
I want to thank the authors for their commitment and dedication to answering my questions. The revised manuscript fully satisfies all questions, and the scientific quality and rigour of the results presented have acquired more robust scientific support.